# A global, regional, and national survey on burden and Quality of Care Index (QCI) of bladder cancer: The global burden of disease study 1990–2019

Amirali Karimi[1‡], Parnian Shobeiri[1‡], Sina Azadnajafabad[1], Masoud Masinaei[1,2], Negar Rezaei[1,3], Ali Ghanbari[1], Nazila Rezaei[1], Mahtab Rouhifard[1], Sarvenaz Shahin[1], Mohammad-Mahdi Rashidi[1], Mohammad Keykhaei[1,4], Ameneh Kazemi[1], Erfan Amini[5], Bagher Larijani[3], Farshad Farzadfar[1,3]*

1 Non-Communicable Diseases Research Center, Endocrinology and Metabolism Population Sciences Institute, Tehran University of Medical Sciences, Tehran, Iran, 2 Department of Epidemiology and Biostatistics, Tehran University of Medical Sciences, Tehran, Iran, 3 Endocrinology and Metabolism Research Center, Endocrinology and Metabolism Clinical Sciences Institute, Tehran University of Medical Sciences, Tehran, Iran, 4 Feinberg Cardiovascular and Renal Research Institute, Northwestern University, School of Medicine, Chicago, United States of America, 5 Uro-Oncology Research Center, Tehran University of Medical Sciences, Tehran, Iran

‡ AK and PS contributed equally to this work and share first authorship on this work.
* f-farzadfar@tums.ac.ir

**Data Availability Statement:** Data Availability Statement: The study protocol and data used in this work are available both from [https://www.

## Abstract

### Introduction

Bladder cancer (BCa) is the second most common genitourinary cancer and among the leading causes of cancer-related deaths. We aimed to assess BCa quality of care (QOC) utilizing a novel multi-variable quality of care index (QCI).

### Materials and methods

Data were retrieved from the Global Burden of Disease 1990–2019 database. QCI scores were calculated using four indices of prevalence-to-incidence ratio, Disability-Adjusted Life Years-to-prevalence ratio, mortality-to-incidence ratio, and Years of Life Lost-to-Years Lived with Disability ratio. We used principal component analysis to allocate 0–100 QCI scores based on region, age groups, year, and gender.

### Results

Global burden of BCa is on the rise with 524,305 (95% UI 475,952–569,434) new BCa cases and 228,735 (95% UI 210743–243193) deaths in 2019, but age-standardized incidence and mortality rates did not increase. Global age-standardized QCI improved from 75.7% in 1990 to 80.9% in 2019. The European and African regions had the highest and lowest age-standardized QCI of 89.7% and 37.6%, respectively. Higher Socio-demographic index (SDI) quintiles had better QCI scores, ranging from 90.1% in high SDI to 30.2% in low

protocols.io/view/quality-of-care-index-qci-bprjmm4n] and the Global Burden of Disease Results Tool [https://ghdx.healthdata.org/gbd-results-tool]. The authors confirm they had no special access or privileges to the data that other researchers would not have.

**Funding:** The author(s) received no specific funding for this work.

**Competing interests:** The authors have declared that no competing interests exist.

**Abbreviations:** BCa, Bladder cancer; DALY, Disability-Adjusted Life Years; GATHER, Guidelines for Accurate and Transparent Health Estimates Reporting; GBD, Global burden of disease; GDR, Gender Disparity Ratio; HAQ, Healthcare Access and Quality; HDI, Human development index; ICD-10, International Classification of Diseases-10; IHME, Institute for Health Metrics and Evaluation; PCA, Principal component analysis; QOC, Quality of care; SCC, Squamous cell carcinoma; SDI, Socio-demographic index; WHO, World Health Organization; YLD, Years Lived with Disability; YLL, Years of Life Lost.

SDI countries in 2019; however, 5-year QCI improvements from 2014 to 2019 were 0.0 for high and 4.7 for low SDI countries.

## Conclusion

The global QCI increased in the last 30 years, but the gender disparities remained relatively unchanged despite substantial improvements in several regions. Higher SDI quintiles had superior QOC and less gender- and age-based inequalities compared to lower SDI countries. We encourage countries to implement the learned lessons and improve their QOC shortcomings.

## Introduction

Bladder cancer (BCa) is the second most common genitourinary cancer and among the leading causes of cancer-related deaths worldwide [1]. BCa accounted for 5% of the total cancer costs in the European Union, with higher-income countries spending higher budgets on this cancer [2]. Besides the strangulating financial burden, BCa imposes a significant toll on the patients' quality of life, either by the tumor behavior or the treatment-related adverse events [3]. Urothelial transitional cell carcinoma is the most common histologic type of BCa, followed by squamous cell carcinoma (SCC) and other less prevalent subtypes, all carrying poorer prognosis compared to the urothelial carcinoma [4,5].

Microscopic or gross hematuria is the most common presentation of BCa. At the time of diagnosis, 70% of the patients have non-muscle invasive disease; but during a 5-year period, 50–70% face tumor recurrence and 10–15% will turn into muscle-invasive disease. The other 30% are diagnosed on the muscle-invasive stage that carry an unfavorable prognosis despite aggressive treatments. In many cases, BCa and its recurrences will require a life-long surveillance. All these sophisticated processes require careful care and a considerable amount of budget, emphasizing the role of a well-designed system for its management.

Earlier studies reported an increased BCa incidence, mortality, and Disability-Adjusted Life Years (DALYs) [6,7]. A higher incidence of BCa requires more resource allocations by the healthcare systems [6]. Higher costs might consequently reduce the availability of health services for patients with BCa. Inequalities in allocating facilities and resources have decelerated the global success on fighting this disease [8–10]. Individuals and countries with lower socio-economic status may suffer more from the complications of BCa compared to higher socio-economic group [8]. Therefore, the Quality of care (QOC) measures should be calculated to analyze and address these inequalities [11]. QOC assures offering skilled and professional services to the patients to achieve the most desirable outcomes [12]. QOC assessment provides essential information on the function of the healthcare systems and visualizes their disparities to allocate quality care to individuals based on their gender, age, and ethnic status [11]. However, previous studies failed to evaluate QOC properly, especially for the BCa [12].

In this study, we aimed to propose the quality of care index (QCI) as a novel multi-variable indicator of QOC. We will compare the QCI scores of various world regions and discuss their age and gender disparities. Visualization of the scores will enable us to discuss the current controversies and hypothesize how to improve QOC. The application of QCI might help healthcare systems establish policies for more proficient and equal resource allocation.

## Material and methods

### 2.1 Design and data resources

We gathered the data from the global burden of disease (GBD) 1990–2019 retrievable from IHME (Institute for Health Metrics and Evaluation) "GBD compare" tool [13]. GBD utilizes the 10th version of International Classification of Diseases (ICD-10) codes C67-C67.9, D09.0, D30.3, D41.4-D41.8, D49.4 for mapping BCa death and C67-C67.9, Z12.6-Z12.79, Z80.52, Z85.51 for mapping BCa new cases (details in **S1 Table**) [14,15]. The burden and the indices related to the QCI of BCa were then calculated. This study is conducted based on the GATHER (Guidelines for Accurate and Transparent Health Estimates Reporting) guidelines [16].

### 2.2 Quality of Care Index (QCI) and its validation

We developed this index to examine the healthcare systems' ability to provide adequate care and access related to diseases. The process and rationale behind QCI development are discussed in the QCI protocol and earlier studies on this index [12,17–20]. QCI constitutes four secondary indices discussed below, selected from six primary candidates. The elements of these indices include prevalence, incidence, DALYs, mortality, Years of Life Lost (YLLs), and Years Lived with Disability (YLDs).

$$Prevalence - to - Incidence \ ratio \ (x) = \frac{Prevalence \ (x)}{Incidence \ (x)} \tag{1}$$

$$Mortality - to - Incidence \ ratio \ (MIR)(x) = \frac{Mortality \ (x)}{Incidence \ (x)} \tag{2}$$

$$DALYs - to - Prevalence \ ratio \ (x) = \frac{DALYs \ (x)}{Prevalence \ (x)} \tag{3}$$

$$YLLs - to - YLDs \ ratio \ (x) = \frac{YLLs \ (x)}{YLDs \ (x)} \tag{4}$$

X represents location (i.e.: countries, regions, or global scale), age group, sex, and year in the above equations.

As a multivariate analysis method, principal component analysis (PCA) is a mathematical approach that combines these abovementioned four secondary indices to create QCI. PCA uses entry datasets and extracts linear combination as orthogonal components [21]. Each of the four mentioned components represents a dimension, and the +200 location data (x) in all the studied 29 years are present in these dimensions. Then PCA uses a four-dimensional transformation of data points (or n-dimensional in other cases) and calculate an eigenvector that covers the highest span on its axis. The eigenvector that best describes the data points' variance and variability, and hence has the highest discrimination capability, will be the first PCA component and considered as a composite characteristic. As mentioned, the first principal component encompasses the highest correlation with the entry variables and provides the best information. Herein, we named this first principal component as QCI and allocated a score of 0–100 to it [17]. Higher QCI scores illustrate a superior QOC for the countries and regions. Illustrations helped visualized the results and strengthened the understanding of QCI. The study by Mohammadi et al. describes the PCA method in more details [17,18].

The socio-demographic index (SDI) combines education, per capita income, and fertility rates to construct an overall development scale for the countries [22,23]. This index categorizes

the countries into five quintiles based on their incomes; high, high-middle, middle, low-middle, and low-income countries [22]. We compared the SDI quintiles to further strengthen our results and discussion;

IHME previously developed Healthcare Access and Quality (HAQ) index [24]. We conducted a mixed effect model to validate QCI by assessing its correlation with the HAQ index. The model applied QCI as a dependent variable, and outpatient care utilization, inpatient care utilization, BCa prevalence, deaths, and attributed death to risk factors as independent variables with countries as the random effects. **S2 Table** demonstrates the details on the results of the conducted mixed effect model for QCI validation. QCI notably correlated with the HAQ index with Pearson's correlation coefficient of 0.80.

## 2.3 Rationale for each QCI component

1. Prevalence-to-incidence ratio: For a specific disease or condition, if the patients receive better QOC, the mortality rates will decrease and a larger portion of the patients will remain alive. Therefore, given a similar incidence, if the QOC is increased, the prevalence of the disease will be higher and prevalence-to-incidence ratio will increase.

2. Mortality-to-incidence ratio (MIR): As mentioned above, given a similar incidence, a worse QOC will yield higher mortality rates and a higher mortality-to-incidence ratio.

3. DALYs-to-prevalence ratio: Similar to MIR, if two regions have similar prevalence rates, the region with a lower QOC will have higher DALYs and consequently, a higher DALYs-to-prevalence ratio.

4. YLLs-to-YLDs: In a case of worse disease management and QOC, patients decease earlier and their YLL increase. On the other hand, YLD will decrease as the patients have died earlier and will live less years with their disabilities. Therefore, lower QOC pertains to a higher YLLs-to-YLDs ratio.

## 2.4 Age and gender disparity

We presented our age groups as five-year intervals starting from 15 to 85+ years (15–19, 20–24, 2 . . ., 80–84, 85 plus). This strategy enabled us to pinpoint any accounts of age disparity for different locations. BCa has a negligible incidence below the age of 15 [13, 25]. We also utilized age-standardized values for the calculation of indices to increase the capabilities of more robust comparisons.

We also introduced the Gender Disparity Ratio (GDR) to analyze the gender-related availability of QOC.

$$GDR\,(x) = \frac{QCI\,(x)\;for\;females}{QCI\,(x)\;for\;males}$$

X represents a location, age group, and year in the above formula. GDR values nearing 1 implied equal QOC among males and females.

## 2.5 Statistical analysis

Age-standardized calculations were based on the GBD world population. We depicted a 95% uncertainty interval (UI) for primary indices. Results were considered significant if the groups did not show UI overlap. PCA method were used to calculate QCI as discussed above. All the

statistical analyses and illustration were carried out using R statistical packages v4.0.4 (http://www.r-project.org/, RRID: SCR_001905).

## Results

### 3.1 Incidence, mortality, and DALYs

**Global.** The number of BCa cases and mortality is on the rise globally. Worldwide, 524,305 (95% UI: 475,952–569,434) new BCa cases and 228,735 (210,743–243,193) deaths were recorded in 2019 compared to 234,754 (225,464–243,075) new cases and 121,500 (114,751–127,171) mortalities in 1990. However, age-standardized incidence rate per 100,000 did not increase in the same period. BCa incidence rate was 6.3 (6.0–6.5) in 1990, 6.5 (6.2–6.7) in 2005, and 6.5 (5.9–7.1) in 2019. There is a decreasing pattern of BCa death and DALYs rate worldwide from 1990 to 2019. BCa death rate is estimated at 3.5 (3.3–3.7), 3.2 (3–3.3), and 2.9 (2.7–3.1) in 1990, 2005, and 2019, respectively. The numbers were 66.6 (63.0–69.7), 60.4 (57.7–62.8), and 54.2 (50.4–58.0) for DALYs rate.

Male patients constituted most of the incident cases, deaths, and DALYs throughout these years. A similar concept was observed in 2019 for the male to female age-standardized rates (incidence: 11.3 vs. 2.7, deaths: 5.1 vs. 1.4, DALYs: 90.2 vs. 24.4) (**Table 1**).

**Regional.** The incidence, mortality, and DALYs of BCa are steadily increasing for all the six World Health Organization (WHO) regions among both genders. European region, Western Pacific region, and the region of the Americas were the main contributors to the incidence and mortality of BCa. European region had 201,655 (178,323–225,564) new cases and 77,923 (70,920–83,278) mortalities in 2019. These numbers were 150,005 (131,044–170,665) and 59,087 (51,799–66,594) for Western Pacific region, and 82,614 (73,277–92,728) and 39,999 (36,462–42,491) for the region of the Americas in the same year. The African region shared the least estimates with 14,946 (12,758–17,128) new cases and 10,808 (9,325–12,274) mortalities. In 2019, European region 12.4 (10.9–13.8), Eastern Mediterranean region 9.3 (7.8–11.3), and the region of the Americas 6.4 (5.7–7.2) had the highest age-standardized incidence rate per 100,000, while South-East Asia region had the lowest incidence rate of 2.1 (1.9–2.4).

### 3.2 QCI

Global age-standardized QCI improved from 68.4% and 75.7% in 1990 and 2005, to 79.8% and 80.9% in 2014 and 2019. Higher SDI quintiles had better QCI scores, ranging from 90.1% in high SDI countries to 30.2% in low SDI countries; however, lower SDI countries showed better 5-year improvements in QCI. The net QCI change was 0.0 for high, 1.1 for high-middle, 4.7 for middle, 5.3 for low-middle, and 4.7 for low SDI countries from 2014 to 2019 (**Table 2**).

In 2019, the top 3 regions were the European region (age-standardized QCI = 89.7%), the Western Pacific region (85.3%), and the region of the Americas (78.4%). The lowest were the African region (37.6%), the South-East Asia region (50.7%), and the Eastern Mediterranean region (71.5%). Italy (99.4%), Australia (98.1%), Iceland (97.5%), Japan (97.2%), and Spain (97.2%) ranked as the top 5; while Central African Republic (6.8%), Somalia (11.0%), South Sudan (14.5%), Guinea (16.8%), and Chad (16.8%) were the bottom 5 countries (**Table 2, Fig 1A**).

**Fig 2** depicts the scatter plot of age-standardized QCI for different countries based on the six WHO world regions. The trend was similar throughout the years, with African region remaining the lowest and European regions owning the highest age-standardized QCI scores during the study period. Eastern Mediterranean region improved more rapidly towards higher SDI and QCI scores compared to the other WHO regions. Similarly, North Africa and Middle East region had the best progress among the seven GBD super-regions (**S1 Fig**). Both figures

**Table 1. Bladder cancer numbers and age-standardized rates (per 100,000) of incidence, mortality, and Disability-Adjusted Life Years (DALYs) at Global and six WHO regions by sex.**

| Location | Sex | 1990 Incidence Number | 1990 Incidence Rate | 1990 Deaths Number | 1990 Deaths Rate | 1990 DALYs Rate | 2005 Incidence Number | 2005 Incidence Rate | 2005 Deaths Number | 2005 Deaths Rate | 2005 DALYs Rate | 2019 Incidence Number | 2019 Incidence Rate | 2019 Deaths Number | 2019 Deaths Rate | 2019 DALYs Rate |
|---|---|---|---|---|---|---|---|---|---|---|---|---|---|---|---|---|
| **Global** | Both | 234,754 (225,464–243,075) | 6.3 (6–6.5) | 121,500 (114,751–127,171) | 3.5 (3.3–3.7) | 66.6 (63–69.7) | 350,187 (335,351–360,998) | 6.5 (6.2–6.7) | 163,613 (154,647–169,690) | 3.2 (3–3.3) | 60.4 (57.7–62.8) | 524,305 (475,952–569,434) | 6.5 (5.9–7.1) | 228,735 (210,743–243,193) | 2.9 (2.7–3.1) | 54.2 (50.4–58) |
|  | Female | 56,992 (53,455–60,307) | 2.8 (2.6–2.9) | 33,376 (30,658–35,843) | 1.7 (1.5–1.8) | 31.7 (29.2–34.3) | 81,888 (75,817–85,654) | 2.8 (2.6–2.9) | 44,063 (40,138–46537) | 1.5 (1.4–1.6) | 27.9 (26.1–29.5) | 116,438 (103,711–128,213) | 2.7 (2.4–2.9) | 59,527 (52,329–64,581) | 1.4 (1.2–1.5) | 24.4 (22.1–26.4) |
|  | Male | 177,762 (170,773–184,132) | 10.9 (10.4–11.3) | 88,124 (83,470–91,864) | 6.1 (5.8–6.4) | 111.7 (105.6–116.6) | 268,299 (259,060–276,156) | 11.2 (10.7–11.6) | 119,550 (114,047–123,576) | 5.6 (5.3–5.8) | 100.6 (96.1–104.3) | 407,866 (371,297–443,750) | 11.3 (10.2–12.3) | 169,207 (156,921–180,655) | 5.1 (4.7–5.4) | 90.2 (83.6–96.6) |
| **African Region** | Both | 6,976 (5,713–8,399) | 3.4 (2.8–4.1) | 5,658 (4,603–6,871) | 3 (2.5–3.7) | 60.8 (49.4–73.9) | 9,765 (8,318–11,652) | 3.3 (2.8–4) | 7,673 (6,460–9,298) | 2.9 (2.4–3.5) | 56.9 (47.9–69) | 14,946 (12,758–17,128) | 3.3 (2.9–3.8) | 10,808 (9,325–12,274) | 2.7 (2.3–3) | 52.5 (45–60.1) |
|  | Female | 2,263 (1,840–3,061) | 2.1 (1.7–2.9) | 1,852 (1,491–2,565) | 1.9 (1.5–2.6) | 38.8 (31.2–52.9) | 3,115 (2,609–4,127) | 2 (1.7–2.7) | 2,508 (2,073–3,441) | 1.8 (1.5–2.4) | 35.2 (29.1–48.1) | 4,549 (3,738–5,264) | 1.9 (1.6–2.2) | 3,492 (2,847–4,028) | 1.6 (1.3–1.8) | 31.3 (25.4–36.4) |
|  | Male | 4,713 (3,735–5,907) | 4.9 (3.9–6.2) | 3,806 (2,990–4,841) | 4.4 (3.5–5.6) | 84.9 (66.8–107.7) | 6,650 (5,549–8,223) | 4.9 (4.1–6.1) | 5,165 (4,213–6,497) | 4.2 (3.5–5.4) | 81.3 (66.5–101.9) | 10,397 (8,585–12,101) | 5.1 (4.2–5.8) | 7,316 (6,104–8,461) | 4 (3.4–4.6) | 77.1 (64–89.3) |
| **Eastern Mediterranean Region** | Both | 11,345 (9,823–12,709) | 6.4 (5.5–7.2) | 7,534 (6,436–8,577) | 4.7 (4–5.4) | 100.5 (86.3–113.5) | 20213 (18790–21758) | 7.8 (7.2–8.3) | 11,415 (10,517–12,360) | 4.9 (4.5–5.4) | 104.2 (96–112.3) | 39,124 (32,304–48,252) | 9.3 (7.8–11.3) | 17,664 (14,880–21,407) | 4.9 (4.1–5.8) | 102.4 (86.2–124.2) |
|  | Female | 2,044 (1,736–2,398) | 2.4 (2–2.9) | 1,498 (1,252–1,779) | 1.9 (1.6–2.3) | 42.2 (35.4–49.8) | 3605 (3260–3950) | 2.9 (2.6–3.2) | 2395 (2124–2663) | 2.1 (1.9–2.4) | 45.3 (40.3–50.2) | 6,521 (5,393–7,890) | 3.3 (2.7–3.9) | 3,718 (3,118–4,398) | 2.1 (1.8–2.5) | 44.1 (36.9–52.3) |
|  | Male | 9,301 (8,004–10,518) | 10 (8.5–11.4) | 6,036 (5,084–6,928) | 7.3 (6–8.4) | 153.3 (130.2–174.9) | 16,609 (15,255–18,085) | 12.3 (11.2–13.4) | 9,019 (8,210–9,901) | 7.5 (6.9–8.3) | 158.5 (144.4–173.4) | 32,602 (26,680–40,762) | 14.9 (12.3–18.3) | 13,946 (11,641–17,149) | 7.4 (6.2–9) | 156.7 (129.9–192.9) |
| **European Region** | Both | 119,679 (115,725–122,720) | 11 (10.7–11.3) | 54,484 (52,146–56,043) | 5.1 (4.8–5.2) | 100.4 (96.8–103.9) | 159,254 (152,653–163,326) | 12.2 (11.7–12.5) | 64,394 (60,680–66,292) | 4.8 (4.6–5) | 94.9 (91.1–97.7) | 201,655 (178,323–225,564) | 12.4 (10.9–13.8) | 77,923 (70,920–83,278) | 4.5 (4.1–4.8) | 84.6 (78.7–90.5) |
|  | Female | 25,303 (23,843–26,177) | 3.8 (3.6–3.9) | 13,529 (12,533–14,072) | 2 (1.8–2.1) | 36.2 (34.4–37.7) | 33,187 (30,718–34,530) | 4.3 (4–4.4) | 15,933 (14,427–16,692) | 1.9 (1.7–2) | 34.5 (32.4–35.8) | 41,735 (36,018–47,132) | 4.4 (3.8–4.9) | 19,063 (16,604–20,767) | 1.8 (1.6–1.9) | 31.6 (28.7–34) |
|  | Male | 94,376 (91,642–96,715) | 22.5 (21.8–23.1) | 40,955 (39,464–42,038) | 10.5 (10.1–10.9) | 199.8 (192.9–206.4) | 126,067 (122,150–129,023) | 23.8 (22.9–24.4) | 48,461 (46,374–49,714) | 9.7 (9.1–10) | 181 (174.3–186.3) | 159,919 (141,017–179,441) | 23.2 (20.5–26) | 58,861 (54,088–62,723) | 8.6 (7.9–9.1) | 155.6 (144.5–166.5) |
| **Region of the Americas** | Both | 40,499 (38,894–41,579) | 6.7 (6.4–6.9) | 20,166 (19,037–20,819) | 3.4 (3.2–3.5) | 64 (61.5–65.9) | 57,907 (55,188–59,484) | 6.7 (6.4–6.8) | 27,976 (26,085–29,041) | 3.2 (3–3.3) | 59.4 (56.8–61.2) | 82,614 (73,277–92,728) | 6.4 (5.7–7.2) | 39,999 (36,462–42,491) | 3.1 (2.8–3.2) | 55.2 (51.6–58.4) |
|  | Female | 12,091 (11,339–12,541) | 3.5 (3.3–3.6) | 6,163 (5,634–6,458) | 1.8 (1.6–1.8) | 32.8 (31–34.1) | 17,565 (16,311–18,260) | 3.6 (3.4–3.7) | 8,842 (7,942–9,307) | 1.7 (1.6–1.8) | 31.9 (30–33.1) | 24,161 (21,068–27,535) | 3.4 (3–3.9) | 11,869 (10,488–12,822) | 1.6 (1.4–1.7) | 29.1 (26.8–31) |
|  | Male | 28,408 (27,501–29,100) | 10.9 (10.5–11.1) | 14,003 (13,422–14,397) | 5.7 (5.4–5.9) | 104.4 (100.8–107.4) | 40,343 (38,841–41,315) | 10.6 (10.2–10.8) | 19,134 (18,102–19,748) | 5.2 (4.9–5.4) | 94 (90.2–96.8) | 58,452 (50,821–66,984) | 10.2 (8.9–11.6) | 28,131 (25,912–29,772) | 5 (4.6–5.3) | 87.3 (81.9–92.2) |

*(Continued)*

**Table 1.** (Continued)

| Location | Sex | 1990 Incidence Number | 1990 Incidence Rate | 1990 Deaths Number | 1990 Deaths Rate | 1990 DALYs Rate | 2005 Incidence Number | 2005 Incidence Rate | 2005 Deaths Number | 2005 Deaths Rate | 2005 DALYs Rate | 2019 Incidence Number | 2019 Incidence Rate | 2019 Deaths Number | 2019 Deaths Rate | 2019 DALYs Rate |
|---|---|---|---|---|---|---|---|---|---|---|---|---|---|---|---|---|
| South-East Asia Region | Both | 11,102 (9,880–12,503) | 1.8 (1.6–2.1) | 8,424 (7,339–9,479) | 1.6 (1.4–1.8) | 30.9 (27–34.7) | 18,521 (16,925–20,266) | 1.9 (1.8–2.1) | 13,130 (11,826–14,411) | 1.6 (1.4–1.7) | 29.3 (26.4–32.2) | 34,474 (30,523–38,738) | 2.1 (1.9–2.4) | 22,524 (20,014–25,161) | 1.5 (1.4–1.7) | 29 (25.7–32.5) |
| | Female | 3,314 (2,648–4,264) | 1.1 (0.9–1.4) | 2,602 (2,068–3,313) | 1 (0.8–1.2) | 19.2 (15.3–24.5) | 5,270 (4,514–6,171) | 1.1 (0.9–1.3) | 3,950 (3,405–4,682) | 0.9 (0.8–1.1) | 17.2 (14.8–20.4) | 9,560 (8,185–10,975) | 1.1 (1–1.3) | 6,727 (5,630–7,774) | 0.9 (0.7–1) | 16.3 (13.6–18.8) |
| | Male | 7,788 (6,878–8,626) | 2.6 (2.3–2.9) | 5,822 (5,007–6,434) | 2.3 (1.9–2.5) | 42.8 (36.8–47.2) | 13,250 (11,964–14,405) | 2.9 (2.6–3.1) | 9,180 (8,153–10,040) | 2.3 (2–2.5) | 42.4 (37.6–46.2) | 24,914 (21,353–28,833) | 3.3 (2.8–3.7) | 15,797 (13,712–18,177) | 2.3 (2–2.6) | 42.9 (37.3–49.4) |
| Western Pacific Region | Both | 44,479 (40,907–48,003) | 4.2 (3.9–4.5) | 24,836 (22,490–27,092) | 2.6 (2–2.8) | 49.3 (44.7–53.9) | 83,517 (77,781–89,117) | 4.9 (4.6–5.2) | 38,504 (35,770–41,394) | 2.5 (2.3–2.7) | 45.8 (42.7–49.4) | 150,005 (131,044–170,665) | 5.7 (5–6.4) | 59,087 (51,799–66,594) | 2.3 (2–2.6) | 42.2 (37–47.7) |
| | Female | 11,788 (10,433–13,267) | 2.1 (1.8–2.3) | 7,613 (6,639–8,678) | 1.4 (1.3–1.6) | 27.5 (23.8–31.6) | 18,882 (16,965–20,432) | 2.1 (1.9–2.3) | 10,284 (9,181–11,168) | 1.2 (1.1–1.3) | 21.4 (19.5–23.3) | 29,555 (24,550–34,688) | 2.1 (1.7–2.4) | 14,466 (11,808–16,760) | 1 (0.8–1.2) | 17.4 (14.8–20.2) |
| | Male | 32,691 (29,728–35,909) | 7.1 (6.5–7.7) | 17,224 (15,265–19,088) | 4.5 (4.1–4.9) | 78 (69.2–86.2) | 64,635 (59,984–69,792) | 8.5 (7.8–9.1) | 28,220 (26,072–30,676) | 4.3 (4–4.7) | 75.8 (70.1–82.1) | 12,0450 (10,3293–14,0495) | 10.1 (8.7–11.7) | 44,621 (38,217–51,554) | 4.3 (3.7–4.8) | 72.5 (62.3–83.5) |

**Table 2. Age-standardized QCI values (%) and GDR of the selected locations in 1990, 2005, 2014, and 2019.**

| Location | 1990 | | | | 2005 | | | | 2014 | | | | 2019 | | | |
|---|---|---|---|---|---|---|---|---|---|---|---|---|---|---|---|---|
| | Both | Female | Male | GDR | Both | Female | Male | GDR | Both | Female | Male | GDR | Both | Female | Male | GDR |
| Global | 68.4 | 63.3 | 69.6 | 0.91 | 75.7 | 70.7 | 77.1 | 0.92 | 79.8 | 74.9 | 81.2 | 0.92 | 80.9 | 76.0 | 82.4 | 0.92 |
| Socio-demographic index (SDI) quintiles | | | | | | | | | | | | | | | | |
| High SDI | 81.6 | 78.9 | 81.8 | 0.96 | 88.1 | 85.8 | 88.6 | 0.97 | 90.1 | 88.6 | 90.5 | 0.98 | 90.1 | 88.9 | 90.6 | 0.98 |
| High-middle SDI | 70.8 | 65.9 | 71.3 | 0.92 | 79.2 | 75.2 | 79.7 | 0.94 | 84.9 | 81.6 | 85.2 | 0.96 | 86.0 | 82.8 | 86.4 | 0.96 |
| Middle SDI | 44.4 | 34.8 | 47.2 | 0.74 | 61.1 | 51.8 | 63.8 | 0.81 | 71.7 | 61.3 | 74.7 | 0.82 | 76.4 | 66.1 | 79.4 | 0.83 |
| Low-middle SDI | 24.7 | 21.1 | 26.7 | 0.79 | 34.5 | 30.2 | 36.8 | 0.82 | 44.7 | 39.4 | 47.2 | 0.83 | 50.0 | 43.9 | 52.8 | 0.83 |
| Low SDI | 12.7 | 9.9 | 14.5 | 0.68 | 17.2 | 16.5 | 17.8 | 0.93 | 25.5 | 23.1 | 27.0 | 0.86 | 30.2 | 27.4 | 31.9 | 0.86 |
| WHO regions | | | | | | | | | | | | | | | | |
| African Region | 21.3 | 17.7 | 23.2 | 0.76 | 26.4 | 21.7 | 28.8 | 0.75 | 32.8 | 26.8 | 35.8 | 0.75 | 37.6 | 31.0 | 40.9 | 0.76 |
| Eastern Mediterranean Region | 43.9 | 32.6 | 47.5 | 0.69 | 58.1 | 43.6 | 62.4 | 0.70 | 66.3 | 51.4 | 70.7 | 0.73 | 71.5 | 56.8 | 75.8 | 0.75 |
| European Region | 78.3 | 74.3 | 78.7 | 0.94 | 85.2 | 82.0 | 85.8 | 0.96 | 89.3 | 87.0 | 89.8 | 0.97 | 89.7 | 87.5 | 90.3 | 0.97 |
| Region of the Americas | 74.0 | 75.7 | 73.3 | 1.03 | 77.4 | 78.1 | 77.4 | 1.01 | 78.0 | 79.5 | 77.7 | 1.02 | 78.4 | 80.1 | 78.2 | 1.02 |
| South-East Asia Region | 27.7 | 21.9 | 30.6 | 0.72 | 38.6 | 32.0 | 41.7 | 0.77 | 46.6 | 40.6 | 49.1 | 0.83 | 50.7 | 44.6 | 53.5 | 0.83 |
| Western Pacific Region | 62.0 | 53.3 | 63.9 | 0.83 | 75.6 | 69.6 | 75.5 | 0.92 | 83.0 | 77.1 | 83.6 | 0.92 | 85.3 | 79.8 | 85.9 | 0.93 |
| Top 5 countries in 2019 with highest age-standardized QCI values | | | | | | | | | | | | | | | | |
| Italy | 91.1 | 89.5 | 91.4 | 0.98 | 98.3 | 97.3 | 98.4 | 0.99 | 100 | 100 | 100 | 1 | 99.4 | 99.6 | 99.6 | 1 |
| Australia | 87.8 | 85.3 | 88.4 | 0.96 | 91.3 | 88.1 | 92.6 | 0.95 | 93.8 | 91.8 | 95.0 | 0.97 | 98.1 | 97.7 | 98.4 | 0.99 |
| Iceland | 89.8 | 87.2 | 91.2 | 0.96 | 95.8 | 94.2 | 96.9 | 0.97 | 97.7 | 95.5 | 98.7 | 0.97 | 97.5 | 96.0 | 98.4 | 0.98 |
| Japan | 89.1 | 84.0 | 90.4 | 0.93 | 94.4 | 90.1 | 95.1 | 0.95 | 96.6 | 92.7 | 97.6 | 0.95 | 97.2 | 93.9 | 98.1 | 0.96 |
| Spain | 86.5 | 79.9 | 87.1 | 0.92 | 94.7 | 88.7 | 95.3 | 0.93 | 97.2 | 93.5 | 97.7 | 0.96 | 97.2 | 93.1 | 97.9 | 0.95 |
| Bottom 5 countries in 2019 with lowest age-standardized QCI values | | | | | | | | | | | | | | | | |
| Chad | 11.6 | 8.3 | 13.9 | 0.60 | 11.8 | 8.5 | 13.9 | 0.61 | 17.0 | 13.5 | 19.2 | 0.70 | 20.5 | 16.8 | 23.0 | 0.73 |
| Guinea | 9.0 | 5.8 | 11.3 | 0.51 | 14.0 | 11.2 | 15.7 | 0.71 | 15.5 | 13.1 | 17.0 | 0.77 | 19.7 | 16.8 | 21.6 | 0.78 |
| South Sudan | 10.2 | 8.0 | 11.6 | 0.69 | 12.2 | 10.9 | 13.7 | 0.80 | 13.1 | 12.3 | 14.5 | 0.85 | 15.6 | 14.5 | 17.2 | 0.84 |
| Somalia | 5.7 | 3.5 | 7.5 | 0.47 | 6.5 | 3.8 | 8.4 | 0.45 | 9.5 | 6.6 | 11.4 | 0.58 | 11.0 | 8.0 | 13.0 | 0.62 |
| Central African Republic | 3.1 | 4.1 | 3.1 | 1.32 | 2.7 | 2.7 | 2.8 | 0.96 | 5.4 | 4.8 | 5.4 | 0.89 | 6.8 | 6.3 | 6.8 | 0.93 |

Year 2014 was added to the previously mentioned years (1990, 2005, and 2019) to find the 5-year differences in the values (2014 to 2019).

demonstrate an almost-linear relationship between the countries' SDI and the observed QCI score, illustrating the indisputable correlation between the two variables. These two figures were depicted for every ten year to demonstrate the changes for each decade.

### 3.3 Gender disparity

Males received better care worldwide compared to females, as the age-standardized GDR stood at 0.92 in 2019. **Fig 1B** illustrates the geographical distribution of gender disparity. This number remained constant throughout 1990–2019, as the Global GDR was 0.91 in 1990 and 0.92 in 2005, 2014. The disparity inversely correlated with the level of SDI. High (GDR = 0.98) and high-middle SDI (GDR = 0.96) quintiles approached 1, while the numbers were 0.83 for middle and low-middle and 0.86 for low SDI groups. No change in the GDR was observed in the last five years for different SDIs, with a minor increase of 0.01 in the GDR of the middle SDI group. However, GDR for the low SDI improved significantly since 1990 with the net change of 0.18, followed by middle SDI quintile that increased 0.09 in the GDR value. The numbers for low-middle, high-middle, and high SDI groups increased 0.04, 0.04, and 0.02, respectively.

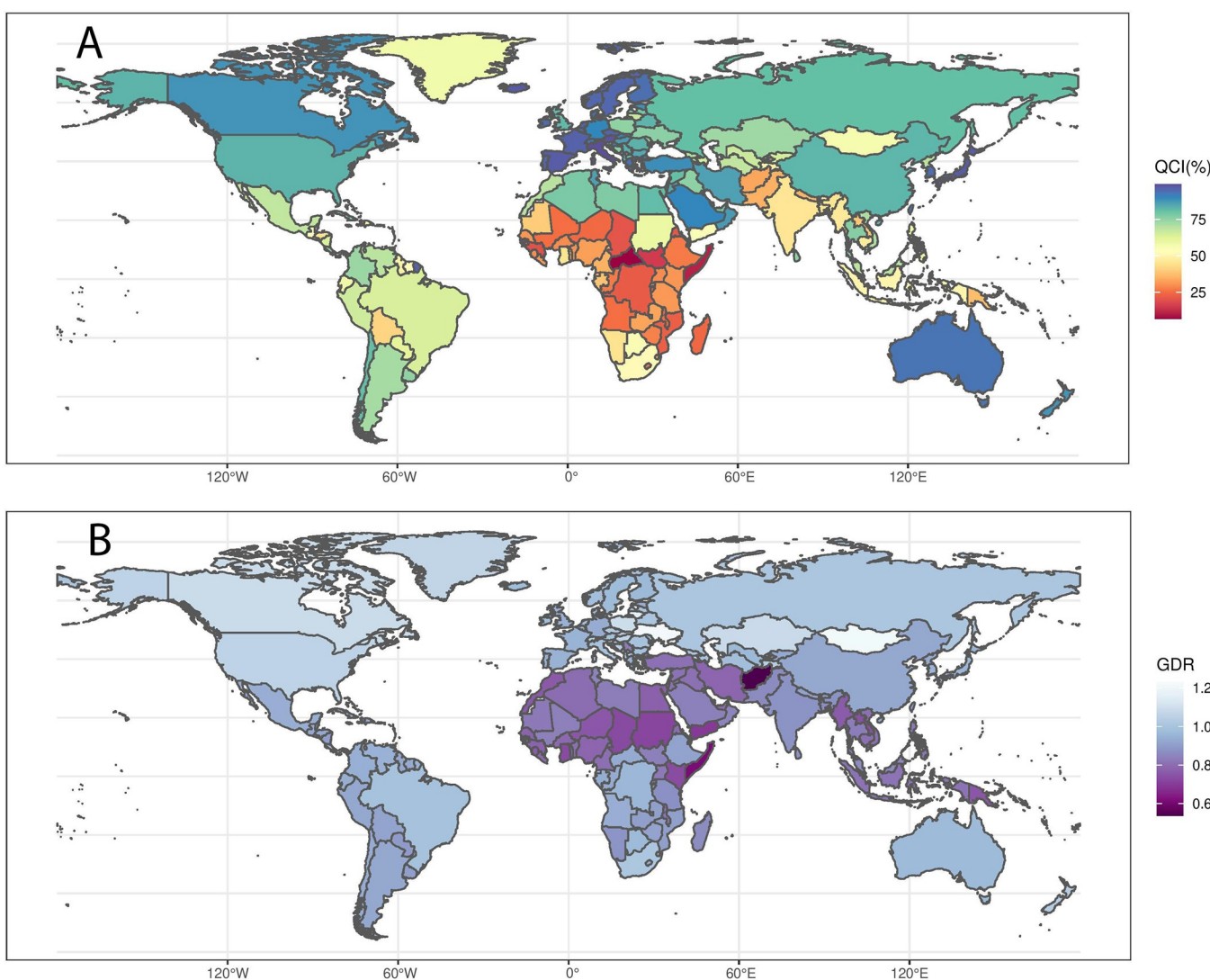

**Fig 1.** A) Age-standardized map of Quality of Care Index (QCI) scores for both genders in 2019 B) Gender Disparity Ratio (GDR) scores in 2019 (Contains information from OpenStreetMap and OpenStreetMap Foundation, which is made available under the Open Database License).

The region of the Americas was the only region with a GDR above 1 (1.02). European region and Western Pacific region followed it with GDR values of 0.97 and 0.93. Eastern Mediterranean region (0.75), the African region (0.75), and the South-East Asia region (0.83) had higher rates of gender disparity. While the South-East Asia region (GDR change: 0.11), Western Pacific region (GDR change: 0.10), and Eastern Mediterranean region (GDR change: 0.06) ameliorated their GDR by more than 0.05 since 1990, the African region showed no progress (GDR change: 0.00) (**Table 2**).

GDR also changed with age groups. GDR was above 1 for all the SDI quintiles before the age of 25 and above 95; however, all these regions showed numbers below 1 in the age groups in between.

### 3.4 Age disparity

Age disparities exist between the age groups in our analysis, evident in **Figs 3 and 4**. The QCI score is readily increasing in both genders and overall and in all the SDI quintiles since

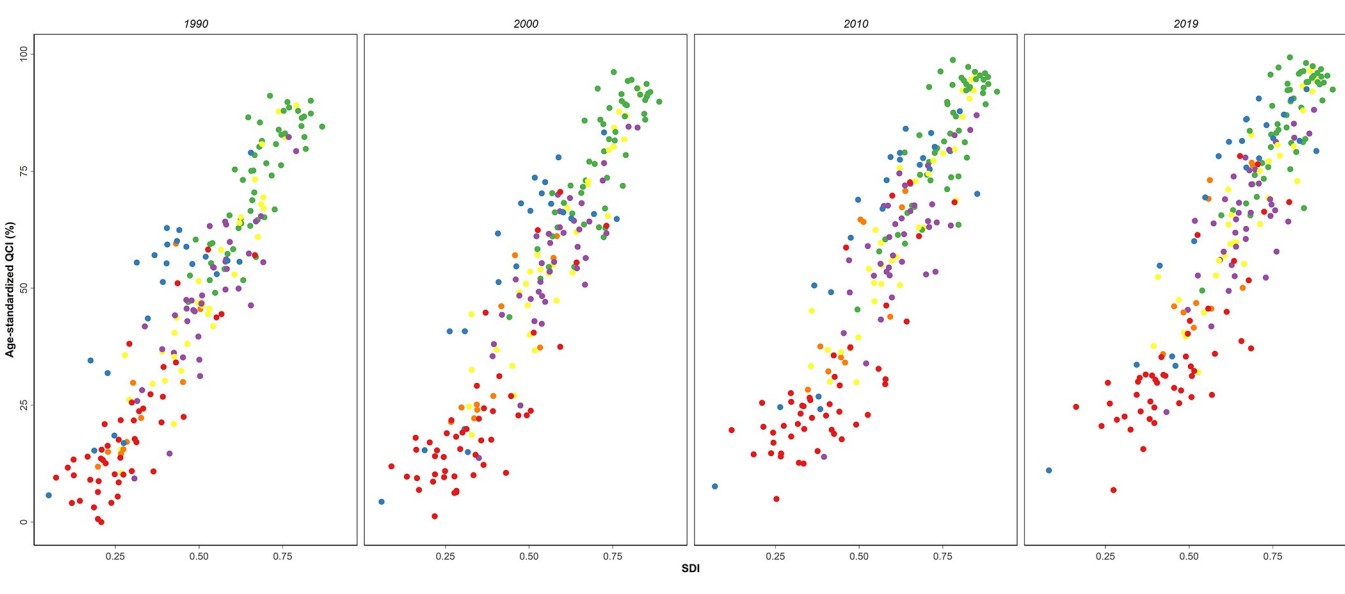

**Fig 2. Scatter plot of age-standardized QCI (%) for the countries by the 6 WHO regions.** Note that the following locations were not initially part of the WHO regions; however, we assigned them to the regions of their locations based on the GBD secretariat opinion. Bermuda, Greenland, and Puerto Rico were assigned to the region of the Americas; Guam, Northern Mariana islands, Taiwan, and Tokelau to the Western Pacific region; and Palestine to the Eastern Mediterranean region.

1990. The global trend resembles more the high and high-middle SDI countries. In 2019, QCI scores of high SDI, high-middle SDI, and overall SDI were above 75 for all the three presented age groups (15–49, 50–75, and 75+) and did not differ much. However, older patients had more remarkably lower SDI in the middle, low-middle, and low SDI countries.

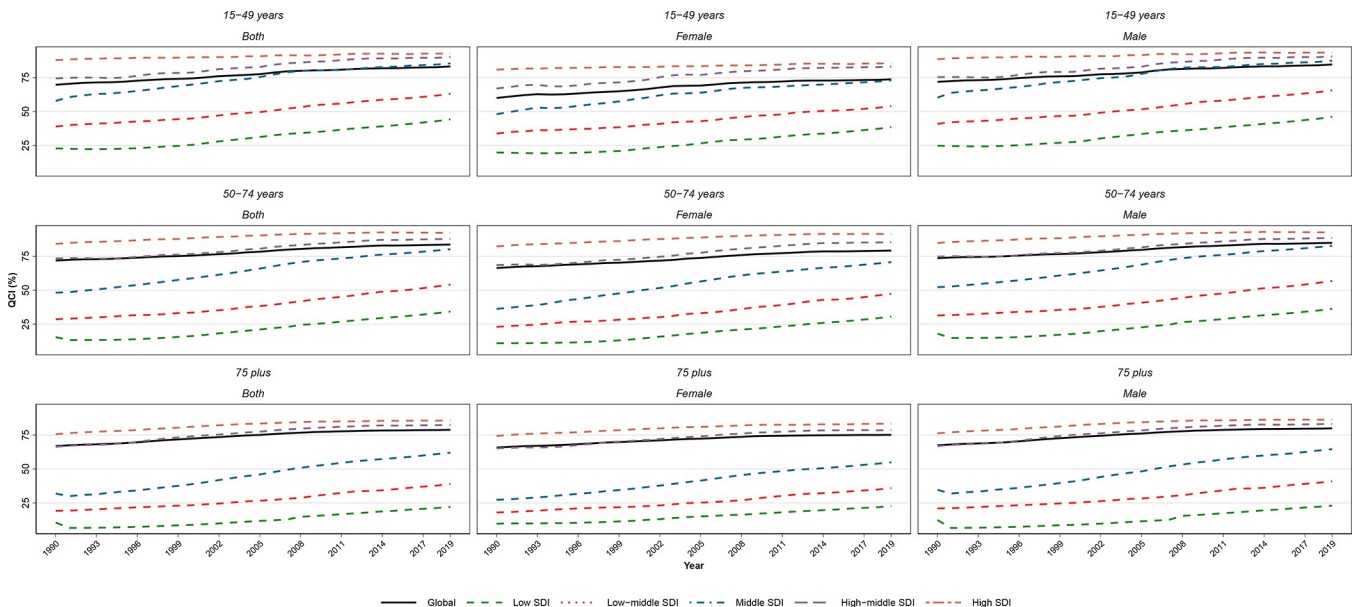

**Fig 3. QCI (%) for age-groups of 15–49, 50–74, and 75+ at Global and Socio-demographic index (SDI) quintiles for both genders 1990–2019.**

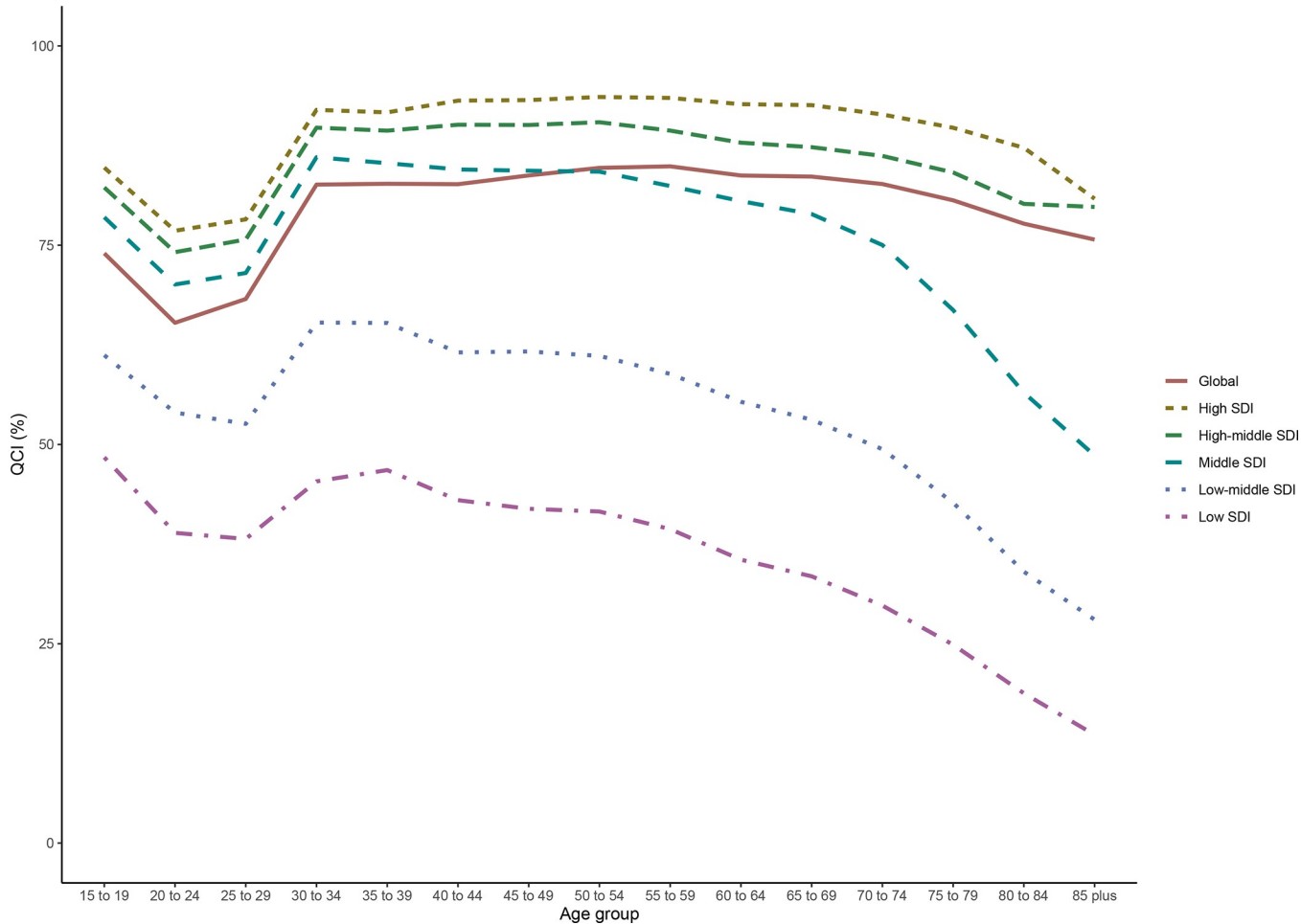

**Fig 4. Age trend of QCI (%) in 5-year categories at Global and Socio-demographic index (SDI) quintiles in 2019.**

In these lower SDI quintiles, 15–49 years had the best QCI scores and 75 plus had the worst numbers.

Fig 4 presents the data for 2019 and the age of the patients are divided into 5-year categories. The overall results showed similar points to Fig 3. In most of the SDI quintiles, QCI scores decline from 15–19 to 20–24, then rise from 20–24 to 30–34, and finally decrease in the 35–39 to 85+ age groups. The 35–85+ years graphs slope differently based on the SDI of the countries. High and high-middle SDI countries experience an approximate plateau for the 35–85+ age group, with a modest decrease in the later parts. On the other hand, middle, low-middle, and low SDI countries demonstrate a steep decline in QCI of 35–85+ years old BCa patients, with an increase in QCI for 35–39 years old patients in low-middle and low SDI countries. Overall, 15–19 years old patients in global (74.0% for 15–19 and 75.7% for 85+), high (84.7% for 15–19 and 80.8% for 85+), and high-middle SDI groups (82.2% for 15–19 and 79.8% for 85+) scored similar to the older patients in 2019. Contrarily, QCI in older BCa patients of the middle (78.5% for 15–19 and 48.6% for 85+), low-middle (61.2% for 15–19 and 28.0% for 85+), and low SDI countries (48.4% for 15–19 and 13.6% for 85+) was substantially lower compared to 15–19 years old group. Middle SDI countries scored better than the global average below 50–54, while they continued to distance from the global QCI after 50 years old.

## Discussion

This presented analysis of GBD 2019 shows the increasing incidence, mortality, and DALYs of BCa. Despite this, the age-standardized rates (per 100,000) of these parameters remained stable worldwide and for most of the regions, while the global age-standardized deaths and DALYs rates significantly decreased. Meantime, age-standardized global QCI is steadily increasing, starting from 68.4% in 1990 and reaching 80.9% in 2019. European region had the highest and African region had the lowest QCI scores throughout the years (2019: 89.7% vs. 37.6%). Gender and age disparities existed throughout the years globally and were more pronounced in the lower SDI quintiles. Global GDR was 0.92 in 2019 and did not improve significantly since 1990 (GDR = 0.91). In 2019, high and low SDI countries had GDRs of 0.98 and 0.85, respectively. Furthermore, patients older than 75 years had gravely lower QCI scores in the middle-to-low SDI quintiles, compared to the youngest BCa patients.

The findings in this study comes concordant to previous GBD analysis regarding the increasing incidence and mortality of BCa, as the world population ages. They found a slight decrease in global age-standardized incidence (1990–2019 change: -5.99%), and a more pronounced reduction in deaths and DALYs (-17.88% and -21.82%, respectively) [6]. While we did not observe a lower incidence rate, our findings confirm the aforementioned decreased age-standardized deaths and DALYs rates. Previous studies utilizing GLOBACAN data from 1990s to 2012 also produce similar results [26,27]. They found stagnant or decreasing incidence rates for men, while incidence in women were on the rise. The death rates, however, were on a decline in most of the countries.

The regional pattern of QCI resembles the findings in the previous studies. In the thyroid, hematologic, neurologic, and breast malignancies, the African region had the lowest QCI scores while the European and Western Pacific regions had the highest scores [12,18,19,28]. The highest and lowest rated countries also had similar patterns, with the highest scoring countries being consistently from higher SDI countries, and the worst QCI scores relating to lower SDI countries.

Gender disparity patterns for BCa was also similar to that of thyroid, hematologic, and neurologic malignancies. In the studies, In the lower SDI quintiles, the GDR values demonstrate lower values, while the higher SDI countries had higher values approaching 1 (equal for both genders) [12,18,19]. Nevertheless, the age disparity patterns are more specific to each cancer. In the aforementioned cancers, the higher SDI quintiles had lower age-based inequalities; however, the target age groups for age disparities differed. Older patients in lower SDI countries had poorer QCI scores compared to younger patients in thyroid cancer [12], similar to our findings on BCa. On the other hand, children were the main targets of inequalities for neurologic malignancies in lower SDI countries [18].

As mentioned earlier, the global burden of BCa is rapidly increasing as the world population ages. Despite the declining rates of tobacco smoking- as the most recognized BCa risk factor- in both genders since 1990 in almost all countries [29], the BCa incidence rate has still remained stagnant. Meanwhile, high and high-middle SDI countries reduced the age-specific incidence rates by controlling the risk factors, while the rates increased in the middle and low-middle countries [6]. The stable global BCa incidence rate- despite the declining smoking rates- might partly relate to the lack of overlap between the regions with the highest BCa prevalence and those with the highest share of smoking. More than 50% of the BCa cases exist in the top 20% human development index (HDI) countries, while only 5% are diagnosed in the low HDI countries [30]. The education and socio-economic status inversely correlate with BCa incidence [31,32]. Moreover, the effect of smoking on the BCa burden is more emphasized in high SDI countries [7]. Meanwhile, nearly 80% of the current smokers live in the low and low-

middle income countries [33]. Furthermore, men share the most cases of BCa with a three-fold incidence compared to women [30], while the middle SDI countries had the highest prevalence of smoking among men [29]. Altogether, countries with higher BCa prevalence are not the ones with greater smoking rates, and this factor can diminish the effect of smoking cessation. On the other hand, efforts failed to successfully assess the changes in the impact of environmental and occupational risk factors of BCa during the past years, such as polycyclic aromatic hydrocarbons, aromatic amines, chlorinated hydrocarbons, and arsenic [34–37]. The progress in the early detection and treatment of the BCa might rationalize the decrease in the observed deaths and DALYs.

The differences in QCI values between SDI quintiles markedly escalate when going down from high to low SDI countries, with the highest gap between the middle and low-middle SDI groups. This increase in gaps demonstrates the growing healthcare access inequalities for BCa based on the countries' level of development. A previous global analysis also confirms that the most developed countries are witnessing the highest decreases in BCa death rates, raising the alarm sign for further inequalities in healthcare access in the future [26].

About 85% of the BCa cases are diagnosed following an episode of painless gross hematuria, and the others present with painless microscopic hematuria [3,38]. Subsequently, the patients will require cystoscopy and other work-ups for diagnosis, followed by costly treatments and regular life-long follow-ups [6]. All these steps impose strangling financial burdens, meaning too much for low-income countries [18,39]. Nevertheless, the numbers in our study might overestimate the QOC deficiencies in African and low SDI countries; as we know, SCC is more common in African countries due to the higher prevalence of Schistosomiasis [40]. As mortality is an important variable in QCI, the poorer prognosis of SCC might drop the QCI numbers in the African countries [5], compared to the other parts of the world where urothelial transitional cell carcinomas are the vast majority of cases.

QCI changes in the last five years are more prominent in lower SDI quintiles, while no 5-year improvement in the QCI score was observed in the high SDI group. This might reflect that the scientific discoveries in the last five years could not increase BCa's QOC and survival, evident by the stagnant QCI in high SDI countries that have quick access to new treatment modalities and innovations. On the other hand, lower SDI quintiles are pushing themselves towards better availability of healthcare access. Furthermore, economic and social developments are changing the epidemiologic pattern of BCa in the African region, from SCC with poorer prognosis to transitional cell carcinoma [41,42]. In recent years, immunotherapeutic agents are capturing the scene and raising hopes for an increased QOC and survival [43]. However, their high cost should raise the alarm of a possible surge in the QOC disparities between the higher and lower SDI quintiles [7].

Some countries with the highest QOC held multi-disciplinary conferences to provide national guidelines and resolve the discrepancies regarding the BCa in their countries [44,45]. An earlier study also found that countries with high QCI scores in thyroid cancer developed national consensus guidelines and constructed cancer registries [12]. These national efforts can strengthen the QOC in the countries and identify and solve the issues specific to each country.

The global age-standardized QCI score comes above the value for the middle SDI countries and runs close to the values for high and high-middle SDI countries, explained by the high prevalence of BCa in developed countries [30–32]. The gap between the global QCI values and middle SDI's QCI gets more pronounced when comparing the patients older than 65. The increased gap for the older patients can be rationalized by the higher life expectancies of the high and high-middle SDI countries compared to the other quintiles [46,47]. Therefore, the

patients' composition, and eventually QCI scores, draw closer to the patients from the higher SDI countries and away from the middle and lower SDI groups in the older ages.

Men are more frequently diagnosed with BCa than women; however, the difference is estimated to narrow down as the percentage of women smoking cigarettes increases [34]. Women suffer from higher mortality to incidence ratio compared to men [48]. This finding comes accordant to our GDR calculations of 0.92 in 2019; however, GDR comes above one below 25 and above 95 years of age. Worse outcomes in women were linked to higher BCa stages and grades with more prevalent multifocal tumors and undesirable histological findings at the time of diagnosis [30,49,50]. Studies also introduced Gender discrimination as a possible contributor to this observation in earlier studies [30,49]. We hypothesize that the gender disparity contributes more to the higher rates of adverse events in women. The GDR approaches 1 in high and high-middle SDI countries with more-widely available healthcare access for both genders, while the number drops to 0.86 and below for other SDI quintiles. The outcomes are superior in higher SDI countries regardless of gender, whereas poorer outcomes are present for women in the lower SDI groups.

Overall, BCa remains a huge healthcare issue with an increasing burden worldwide with little survival improvements in recent years [51]. Urologic malignancies- among them BCa- were among the worst cancers in patient satisfaction [52]. Healthcare policymakers should implement due programs to combat the associated adverse events. Screening the asymptomatic individuals has no proven benefits in the studies and should not be recommended routinely [53], especially for the low-income countries where budget allocations have to be directed towards the most cost-beneficial actions. Therefore, addressing the risk factors remains an important step to decrease the burden of BCa. Reducing tobacco smoking is the most-effective intervention to prevent BCa [54], and healthcare systems should focus the most on this behavior. Secondly, countries at higher risk should reduce the environmental and occupation exposure, and maximize the protection of the people largely exposed to the pollutants [34–37]. Thirdly, endemic regions should implement specific measures to diminish schistosomiasis rates, such as improving sanity and changing behavioral patterns [41].

This study comes with some limitations. First, we could not estimate ethnic and racial inequalities due to the lack of GBD data. Second, IHME-GBD databases might have imprecise or deficient registries for some of the countries. Third, the GBD database did not dichotomize BCa into muscle-invasive and non-muscle-invasive subtypes, while analyzing the data on muscle-invasive BCa comes with great benefits, as it is responsible for a large proportion of BCa-related deaths and DALYs. To our knowledge, this study provides the first comprehensive estimates of BCa QOC using QCI as a validated parameter. Despite all the mentioned limitations, QCI calculations extracted numerous invaluable information requiring specific attention by the international community. Countries can learn from the potentials generated by this score to re-organize their healthcare systems and provide quality care to BCa patients.

## Conclusion

This study introduces the first calculations of QCI and the related age and gender disparities for BCa. The global QCI steadily increased in the last 30 years, but the gender disparities remained relatively unchanged despite substantial improvements in several regions. High and high-middle SDI countries had superior QOC and less gender- and age-based inequalities compared to lower SDI countries. Middle-to-low SDI quintiles have lower QCI scores compared to higher quintiles; specifically, older patients in the lower quintiles receive less quality care and should be emphasized more in the healthcare programs. We recommend the proficient healthcare systems publish more information on their strategies to support BCa patients.

We encourage follow-up studies to evaluate the countries' progress and offer precious findings for healthcare policy makers.

## Supporting information

**S1 Fig. Scatter plot of age-standardized QCI (%) for the countries by the 7 GBD super-regions.**
(PDF)

**S1 Table. ICD-10 codes of bladder cancer mapped for GBD 2019.**
(PDF)

**S2 Table. The coefficients of the mixed-effect regression model.**
(DOCX)

## Acknowledgments

We profoundly thank all staff and colleagues in Non-Communicable Diseases Research Center (NCDRC) and Endocrinology and Metabolism Research Institute (EMRI) at Tehran University of Medical Sciences, helping conducting such valuable studies.

## Author Contributions

**Conceptualization:** Bagher Larijani, Farshad Farzadfar.

**Data curation:** Parnian Shobeiri, Sina Azadnajafabad, Masoud Masinaei, Negar Rezaei, Ali Ghanbari, Mahtab Rouhifard, Sarvenaz Shahin, Mohammad-Mahdi Rashidi, Mohammad Keykhaei.

**Formal analysis:** Amirali Karimi, Parnian Shobeiri, Masoud Masinaei, Negar Rezaei, Ali Ghanbari.

**Investigation:** Masoud Masinaei, Negar Rezaei, Ali Ghanbari, Nazila Rezaei, Ameneh Kazemi, Bagher Larijani.

**Methodology:** Amirali Karimi.

**Project administration:** Nazila Rezaei.

**Software:** Erfan Amini.

**Supervision:** Bagher Larijani, Farshad Farzadfar.

**Validation:** Sina Azadnajafabad, Mahtab Rouhifard, Sarvenaz Shahin, Mohammad-Mahdi Rashidi, Mohammad Keykhaei, Erfan Amini.

**Visualization:** Parnian Shobeiri.

**Writing – original draft:** Amirali Karimi, Parnian Shobeiri.

**Writing – review & editing:** Ali Ghanbari, Nazila Rezaei, Ameneh Kazemi, Erfan Amini, Bagher Larijani, Farshad Farzadfar.

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
