## [Decision Letter · Decision Letter 0]

7 Jun 2022

PONE-D-22-07813A global, regional, and national survey on burden and Quality of Care Index (QCI) of bladder cancer: the global burden of disease study 1990-2019PLOS ONE

Dear Prof. Farshad Farzadfar,

Thank you for submitting your manuscript to PLOS ONE. After careful consideration, we feel that it has merit but does not fully meet PLOS ONE’s publication criteria as it currently stands. Therefore, we invite you to submit a revised version of the manuscript that addresses the points raised during the review process.

We look forward to receiving your revised manuscript.

Kind regards,

Deepak Dhamnetiya, MD

Academic Editor

PLOS ONE

Journal Requirements:

2. We note that Figure 1 in your submission contain [map/satellite] images which may be copyrighted. All PLOS content is published under the Creative Commons Attribution License (CC BY 4.0), which means that the manuscript, images, and Supporting Information files will be freely available online, and any third party is permitted to access, download, copy, distribute, and use these materials in any way, even commercially, with proper attribution. For these reasons, we cannot publish previously copyrighted maps or satellite images created using proprietary data, such as Google software (Google Maps, Street View, and Earth). For more information, see our copyright guidelines: http://journals.plos.org/plosone/s/licenses-and-copyright.

Reviewers' comments:

Reviewer's Responses to Questions

**Comments to the Author**

1. Is the manuscript technically sound, and do the data support the conclusions?

Reviewer #1: Yes

Reviewer #2: Yes

2. Has the statistical analysis been performed appropriately and rigorously? 

Reviewer #1: Yes

Reviewer #2: Yes

3. Have the authors made all data underlying the findings in their manuscript fully available?

Reviewer #1: Yes

Reviewer #2: No

4. Is the manuscript presented in an intelligible fashion and written in standard English?

Reviewer #1: Yes

Reviewer #2: Yes

5. Review Comments to the Author

Reviewer #1: Comment: Thanks to the authors for their hard work on this study “A global, regional, and national survey on burden and Quality of Care Index (QCI) of bladder cancer: the global burden of disease study 1990-2019”. Overall, the article is well-written and has a relevant emphasis. These recommendations and comments, however, may help to enhance this study enough to get it published.

Comment-1: The article has no page number and no line number making it difficult to reference texts for revision.

Comment-2: After reading the introduction, I think it would have been beneficial to have a little more information there.

Comment-3: Please elaborate more about PCA method.

Comment-4: Try to make the discussion more focused on key variable.

Reviewer #2: Congratulations for your effort to represent about such data of regionally and globally concerned. Analysis is worked well and you can discuss comprehensively according to you findings. In addition to these, the following comments are recommended.

1.Firstly, authors should prepare the manuscript according to the PLOS ONE guidelines that include to add number line by line. It is useful for reviewers to give comments to authors. So may I highlight my comments as subtitles or page number in spite of line number.

2. Author cited the Table 1 and Table 2 in the text for the findings but they were not found as neither main table nor supplementary table. The Table 1 you attached as supplementary information is for ICD-10Code.

3.For material and methods, mixed effect model for QCI validation is good to show as the supplementary table or figure.

4. In results section 3.1, findings on national level should be included to reflect the Title of manuscript.

Moreover, all indices of PCA are necessary to be described because the author showed incidence, mortality and DALYs and left over about prevalence, YLLs and YLDs.

5. For section 3.1, corresponding table or figures will make clear presentation about the findings.

6. For section 3.2 QCI, the reason of analysis for only 1990, 2005,2014 and 2019 should be mentioned (refers to the first line of the first paragraph)

7.For Age disparity, age interval are not equally described where they are (a) 3.4, first paragraph, third line , 50-75 and (b) 3.4, second paragraph, second line, 30-43

8. Age group should be included in text for high and high middle SDI countries in place of writing as "above mentioned age group and in the later part" for easy understanding of the readers. (3.4, second paragraph, third to fifth line)

9. The purpose of supplementary figure 1 should be mentioned and what its difference from figure 1 is wanted to know.

10. The year specification as 1990, 2000, 2010 and 2019 for these above figures should be explained for better presentation.

6. PLOS authors have the option to publish the peer review history of their article (what does this mean?). If published, this will include your full peer review and any attached files.

Reviewer #1: No

Reviewer #2: No

---

## [Author Response · Author response to Decision Letter 0]

10 Sep 2022

A global, regional, and national survey on burden and Quality of Care Index (QCI) of bladder cancer: the global burden of disease study 1990-2019

Editor's Comments to Author:

Dear Prof. Farshad Farzadfar,

Thank you for submitting your manuscript to PLOS ONE. After careful consideration, we feel that it has merit but does not fully meet PLOS ONE’s publication criteria as it currently stands. Therefore, we invite you to submit a revised version of the manuscript that addresses the points raised during the review process.

We look forward to receiving your revised manuscript.

Kind regards,

Deepak Dhamnetiya, MD

Academic Editor

PLOS ONE

Journal Requirements:

2. We note that Figure 1 in your submission contain [map/satellite] images which may be copyrighted. All PLOS content is published under the Creative Commons Attribution License (CC BY 4.0), which means that the manuscript, images, and Supporting Information files will be freely available online, and any third party is permitted to access, download, copy, distribute, and use these materials in any way, even commercially, with proper attribution. For these reasons, we cannot publish previously copyrighted maps or satellite images created using proprietary data, such as Google software (Google Maps, Street View, and Earth). For more information, see our copyright guidelines: http://journals.plos.org/plosone/s/licenses-and-copyright.

Answer: Thanks for your suggestions. We revised the article according to the journal style. Furthermore, we added the license statement for Figure 1.

We have revised the manuscript according to the journal instructions and thoughtful suggestions of the reviewers and all the issues were addressed in response to the reviewers.

Reviewer(s)' Comments to Author:

Reviewer: 1

Comments to the Author

Thanks to the authors for their hard work on this study “A global, regional, and national survey on burden and Quality of Care Index (QCI) of bladder cancer: the global burden of disease study 1990-2019”. Overall, the article is well-written and has a relevant emphasis. These recommendations and comments, however, may help to enhance this study enough to get it published.

Q1: The article has no page number and no line number making it difficult to reference texts for revision.

A1: Many thanks for the kind comments. We added the page and line numbers, and also edited the manuscript thoroughly to fit with the journal format accordingly.

Q2: After reading the introduction, I think it would have been beneficial to have a little more information there.

A2: Many thanks. We added a complete paragraph to the introduction accordingly.

Q3: Please elaborate more about PCA method.

A3: Many thanks. We added several more sentences to the methods describing this method and highlighted them accordingly. 

Q4: Try to make the discussion more focused on key variable.

A4: Many thanks for the instructive comment. We went through several important studies related to our manuscript and added their data to our study to strengthen our discussion accordingly.

Reviewer: 2

Congratulations for your effort to represent about such data of regionally and globally concerned. Analysis is worked well and you can discuss comprehensively according to your findings. In addition to these, the following comments are recommended.

Q1: Firstly, authors should prepare the manuscript according to the PLOS ONE guidelines that include to add number line by line. It is useful for reviewers to give comments to authors. So may I highlight my comments as subtitles or page number in spite of line number.

A1: Many thanks for your kind words. We added the page and line numbers and completely revised the manuscript according the Plos one guidelines.

Q2: Author cited the Table 1 and Table 2 in the text for the findings but they were not found as neither main table nor supplementary table. The Table 1 you attached as supplementary information is for ICD-10Code.

A2: Many thanks. The tables were submitted as separate files, but we added them to the text in the revisions according the journal style, and can be visible now. Supplementary table 1 is about ICD-10 codes, but table 1 and table 2 present important data regarding our study.

Q3: For material and methods, mixed effect model for QCI validation is good to show as the supplementary table or figure.

A3: Many thanks. We added the requested table as supplementary table 2 (S2 Table) accordingly.

Q4: In results section 3.1, findings on national level should be included to reflect the Title of manuscript.

Moreover, all indices of PCA are necessary to be described because the author showed incidence, mortality and DALYs and left over about prevalence, YLLs and YLDs.

A4: Many thanks. 

The main aim of this study is QCI that combines all those mentioned variables using PCA. Section 3.1 starts as an introduction to the results section that makes the readers familiar with the epidemiologic situation of BCa, the main novelty of the study is related to the QCI that contains more extensive results, including those of national level. Therefore, we tried not to make this section over-extended, and only tried to focus on the main indices of mortality, incidence, and DALY.

Q5: For section 3.1, corresponding table or figures will make clear presentation about the findings.

A5: Many thanks. Table 1 is added to the text that extensively presents the findings accordingly.

Q6: For section 3.2 QCI, the reason of analysis for only 1990, 2005,2014 and 2019 should be mentioned (refers to the first line of the first paragraph)

A6: Many thanks. The main years in the study were 1990 and 2019, and we also added the year between these values. We added the year 2014 to the analysis to find the 5-year differences. We added the reasoning to the Table 2 footnote (corresponding to section 3.2) accordingly.

Q7: For Age disparity, age interval are not equally described where they are (a) 3.4, first paragraph, third line , 50-75 and (b) 3.4, second paragraph, second line, 30-43

A7: Many thanks for the precise comment. The 30-43 number was wrong, the right number was 30-34 and was corrected accordingly.

The first paragraph describes 3 overall age groups (Figure 3) (according to the age groups of GBD data), while the second paragraph describes ages by 5-year intervals (Figure 4).

Q8: Age group should be included in text for high and high middle SDI countries in place of writing as "above mentioned age group and in the later part" for easy understanding of the readers. (3.4, second paragraph, third to fifth line)

A8: Many thanks. Revised accordingly.

Q9: The purpose of supplementary figure 1 should be mentioned and what its difference from figure 1 is wanted to know.

A9: Many thanks. Supplementary figure 1 demonstrates all the countries in a manner that their region of origin is available. Furthermore, Supplementary Figure 1 also illustrates the situation in various years. Therefore, as we added to the text (highlighted accordingly), this figure facilitates the interpretation of the regions’ progress during the years. On the other hand, Figure 1 shows the situation of the QCI scores and GDR (gender disparity ratio) in specific countries using a word map.

The most similar figure to Supplementary figure 1 is figure 2 that we believe is what the respected reviewer meant. The difference is that figure 2 is related to 6 “WHO” regions, Supplementary figure 1 is related to “GBD” super-regions. We clarified this matter and highlighted accordingly.

Q10: The year specification as 1990, 2000, 2010 and 2019 for these above figures should be explained for better presentation.

A10: Many thanks. Added to the text and highlighted accordingly.

---

## [Decision Letter · Decision Letter 1]

21 Sep 2022

A global, regional, and national survey on burden and Quality of Care Index (QCI) of bladder cancer: the global burden of disease study 1990-2019

PONE-D-22-07813R1

Dear Dr. Farshad Farzadfar,

We’re pleased to inform you that your manuscript has been judged scientifically suitable for publication and will be formally accepted for publication once it meets all outstanding technical requirements.

Kind regards,

Deepak Dhamnetiya, MD

Academic Editor

PLOS ONE

Reviewers' comments:

Reviewer's Responses to Questions

**Comments to the Author**

1. If the authors have adequately addressed your comments raised in a previous round of review and you feel that this manuscript is now acceptable for publication, you may indicate that here to bypass the “Comments to the Author” section, enter your conflict of interest statement in the “Confidential to Editor” section, and submit your "Accept" recommendation.

Reviewer #1: All comments have been addressed

Reviewer #2: All comments have been addressed

2. Is the manuscript technically sound, and do the data support the conclusions?

Reviewer #1: Yes

Reviewer #2: Yes

3. Has the statistical analysis been performed appropriately and rigorously? 

Reviewer #1: Yes

Reviewer #2: Yes

4. Have the authors made all data underlying the findings in their manuscript fully available?

Reviewer #1: Yes

Reviewer #2: No

5. Is the manuscript presented in an intelligible fashion and written in standard English?

Reviewer #1: Yes

Reviewer #2: Yes

6. Review Comments to the Author

Reviewer #1: (No Response)

Reviewer #2: Thank you so much for your detailed reply for my previous comments and congratulations for your great efforts. Your manuscript is acceptable for publication but I am still interested to the data for some QCI constitutes; Prevalence, YLLs and YLDs. If you are feasible, may I suggest to add these data as supporting information for your perfect publication.

7. PLOS authors have the option to publish the peer review history of their article (what does this mean?). If published, this will include your full peer review and any attached files.

Reviewer #1: No

Reviewer #2: **Yes: **Dr. May Soe Aung, Associate Professor, University of Medicine (1), Yangon, Myanmar

---

## [Editor Report · Acceptance letter]

10 Oct 2022

PONE-D-22-07813R1 

A global, regional, and national survey on burden and Quality of Care Index (QCI) of bladder cancer: the global burden of disease study 1990-2019 

Dear Dr. Farzadfar:

I'm pleased to inform you that your manuscript has been deemed suitable for publication in PLOS ONE. Congratulations! Your manuscript is now with our production department. 

Kind regards, 

on behalf of

Dr. Deepak Dhamnetiya 

Academic Editor

PLOS ONE